**DOI: 10.1038/ncomms14737**　　**OPEN**

# Control of type III protein secretion using a minimal genetic system

Miryoung Song[1], David J. Sukovich[1], Luciano Ciccarelli[2,3,4,5], Julia Mayr[2,3,4,5], Jesus Fernandez-Rodriguez[1], Ethan A. Mirsky[1], Alex C. Tucker[1], D. Benjamin Gordon[6], Thomas C. Marlovits[2,3,4,5] & Christopher A. Voigt[1]

Gram-negative bacteria secrete proteins using a type III secretion system (T3SS), which functions as a needle-like molecular machine. The many proteins involved in T3SS construction are tightly regulated due to its role in pathogenesis and motility. Here, starting with the 35 kb *Salmonella* pathogenicity island 1 (SPI-1), we eliminated internal regulation and simplified the genetics by removing or recoding genes, scrambling gene order and replacing all non-coding DNA with synthetic genetic parts. This process results in a 16 kb cluster that shares no sequence identity, regulation or organizational principles with SPI-1. Building this simplified system led to the discovery of essential roles for an internal start site (SpaO) and small RNA (InvR). Further, it can be controlled using synthetic regulatory circuits, including under SPI-1 repressing conditions. This work reveals an incredible post-transcriptional robustness in T3SS assembly and aids its control as a tool in biotechnology.

[1] Department of Biological Engineering, Massachusetts Institute of Technology, Synthetic Biology Center, Cambridge, Massachusetts 02139, USA. [2] Center for Structural Systems Biology (CSSB), University Medical Center Hamburg-Eppendorf (UKE), Martinistrasse 52, D-20246 Hamburg, Germany. [3] Deutsches Elektronen-Synchrotron Zentrum (DESY), Notkestrasse 85, D-22607 Hamburg, Germany. [4] Institute of Molecular Biotechnology of the Austrian Academy of Sciences (IMBA), Vienna Biocenter (VBC), Dr. Bohr-Gasse 3, 1030 Vienna, Austria. [5] Research Institute of Molecular Pathology (IMP), Vienna Biocenter (VBC), Campus-Vienna-Biocenter 1, 1030 Vienna, Austria. [6] MIT-Broad Foundry, Broad Institute of MIT and Harvard, Cambridge, Massachusetts 02139, USA. Correspondence and requests for materials should be addressed to C.A.V. (email: cavoigt@gmail.com).

Bacteria use T3SSs to export proteins through both the inner and outer membranes. They have a central role in virulence for many human pathogens (e.g., *Salmonella, Yersinia, Shigella, Escherichia coli*) and in building the flagellum for motility[1–3]. It has also been harnessed in biotechnology for the production of materials and biologics, agricultural biocontrol agents, cancer therapies and vaccine delivery[4–9]. The T3SS forms a large needle (3.5 MDa) with rings in the inner and outer membranes and a secretion tunnel that extends 50 nm from the cell surface[10,11]. The assembly of the structure is hierarchical, where substructures nucleate first and then are built upon to form the needle complex[12–14]. T3SS genes are under strict regulatory control because the needle is metabolically expensive, they are transiently required during pathogenesis and they are coordinated with other membrane-associated structures[15–17].

One of the most well-studied T3SSs is encoded in *Salmonella* pathogenicity island 1 (SPI-1) and is involved in the invasion of epithelial cells during an early stage of infection[18,19]. In addition to the T3SS genes, SPI-1 also encodes effectors and chaperones[20,21]. It also contains an internal regulatory network that is embedded in global regulation[22]. This network integrates many signals and consists of coupled feedforward and feedback loops[23,24]. Many regulatory motifs are also encoded in the genetics of SPI-1, including regulatory 5′- and 3′-UTRs, long promoters with many operators and start sites, transcriptional interference, transcriptional and translational control internal to genes, translational coupling, and sensitivity to DNA supercoiling[25–29].

Collectively, the overlapping, redundant and non-modular organization of natural genetics complicates the interpretation of top-down genetic changes because a single change can lead to a web of effects. Even when regulation appears to be related to a functional need, proving the linkage is difficult. For example, in distinguishing whether gene order and operons are required for efficient T3SS assembly[30,31] or are a superfluous remnant of evolution[32,33]. Here, we have taken the bottom-up approach of rebuilding SPI-1 from scratch, only retaining those genes and organizational principles required for T3SS assembly and function.

The goal is to create an alternative 'refactored' genetic encoding[34] where the genetics are modular, each part is fully characterized, and whose induction can be controlled with a synthetic circuit (Fig. 1b)[35]. Refactoring involves the elimination of non-coding DNA, internal transcription factors and non-essential genes. Regulation is eliminated from open reading frames by selecting new codons to diversify the underlying DNA sequence. Genes are organized into artificial operons and expression is controlled with synthetic promoters, 5′-UTRs, ribosome-binding sites (RBSs) and terminators. A separate 'controller' is built that contains genetic sensors and circuits, the output of which expresses the phage polymerase to induce expression of T3SS genes. Many intermediate designs failed and the debugging process revealed which features are disposable, as well as identified new essential regulation. During this process, electron microscopy (EM) is used to visualize the structure of needle complexes and this proves critical in determining how part choices impact that the final assembled structures.

The end result of this work is an ultrasimplified 16 kb gene cluster that contains the minimal genetics required for assembly and function. This sharpens our understanding of which conserved regulatory mechanisms are required to build this sophisticated molecular machine. Further, it simplifies its use in biotechnology by decoupling the system from the greater regulatory network such that it can be controlled using synthetic regulatory circuits[38].

## Results

**Overview of refactoring process.** The complete refactored SPI-1 is shown in Figure 1b and the sequence of each component part is shown in Supplementary Table 6. Four intermediate steps were taken to get to this final cluster. First, the *prgHJIKorgABC* genes were refactored together and tested for complementation in a Salmonella typimurium ?prgH-orgC strain (Supplementary Note 2.B). These genes make up the inner membrane ring, needle filament, and portions of the sorting platform that loads proteins from the cytoplasm to the needle[3,21]. Second, the invHFGEACIJspaOPQRS genes were refactored and tested in a *ΔinvH-spaS* strain. These genes include those that comprise outer membrane ring and sorting platform[3,21,37]. Then, the two refactored operons were tested together in the double knockout *ΔprgH-orgC?invH-spaS*, which retains all of the intervening SPI-1 genes, including effectors, chaperones, peptidoglycan/LPS modifying enzymes, and regulatory proteins (Figure 1a). To differentially control the two refactored operons, we chose to use two phage RNAPs that do not cross-react with each other s promoters[38]. Finally, the complete refactored system, including controller, was tested in a ΔSPI-1 strain where the entire cluster is knocked out. At each step, the initial designs were suboptimal or non-functional and had to be fixed through rounds of debugging, described below.

**Refactoring the *prg-org* operon.** The *prgH-orgC* genes were refactored and synthesized as a complete operon, and tested in a Δ*prgH-orgC* knockout. The *prgHIJK* and *orgAB* genes were recoded and only share 38% codon identity to the wild-type sequences. OrgC had been shown to be non-essential and was not included[39]. A library of synthetic RBSs was quantified in *Salmonella* and these were used to control the expression of each gene, the desired levels of which were estimated from the ratios observed in a supramolecular assembly[10] and the experimental characterization of the native RBS strengths (Supplementary Note 1A and Supplementary Fig. 2). Those genes requiring the highest expression were placed at the 5′-end of the operon; this leads to a permuted order of *prgIJHKorgAB* (Supplementary Note 1A). This and the modular design disrupt four regions of translational coupling (Fig. 1a). The resulting 3.9 kb operon was placed under IPTG-inducible $P_{tac}$ control on a pSC101 plasmid backbone. A secretion assay was used to detect FLAG-tagged SptP effector in the supernatant as well as the presence of GroEL as a control for non-specific leakage (see Methods). The Δ*prgH-orgC* knockout can be complemented by a plasmid containing the native operon (including the *prgH* promoter) or the refactored version (0.1 mM IPTG) (Fig. 2a).

Because the refactored *prgH-orgC* operon was able to complement secretion, we progressed to the second SPI-1 operon (below). However, when we later imaged the needles using EM after the *invH-SpaS* operon was refactored (Fig. 1b), we found that they lacked extracellular needle protrusions (Fig. 2b). Because PrgI forms the needle filament and PrgJ has been implicated in needle length[10,40], we decided to increase their expression. Since we already used a strong RBS for *prgI*, we added a synthetic 5′-UTR that contains a hairpin and ribozyme (RiboJ) that we have observed increases expression[41]. Indeed, this increases the expression of PrgI and PrgJ, leading to structures with the correct needle length (Fig. 2c and Supplementary Fig. 9).

**Refactoring the *inv-spa* operon.** A similar approach was taken to design the refactored *invH-spaS* operon. Less information about protein ratios could be gleaned from supramolecular assembly, so there was more reliance on the native RBS strengths, with the default of keeping the levels approximately equal (Supplementary

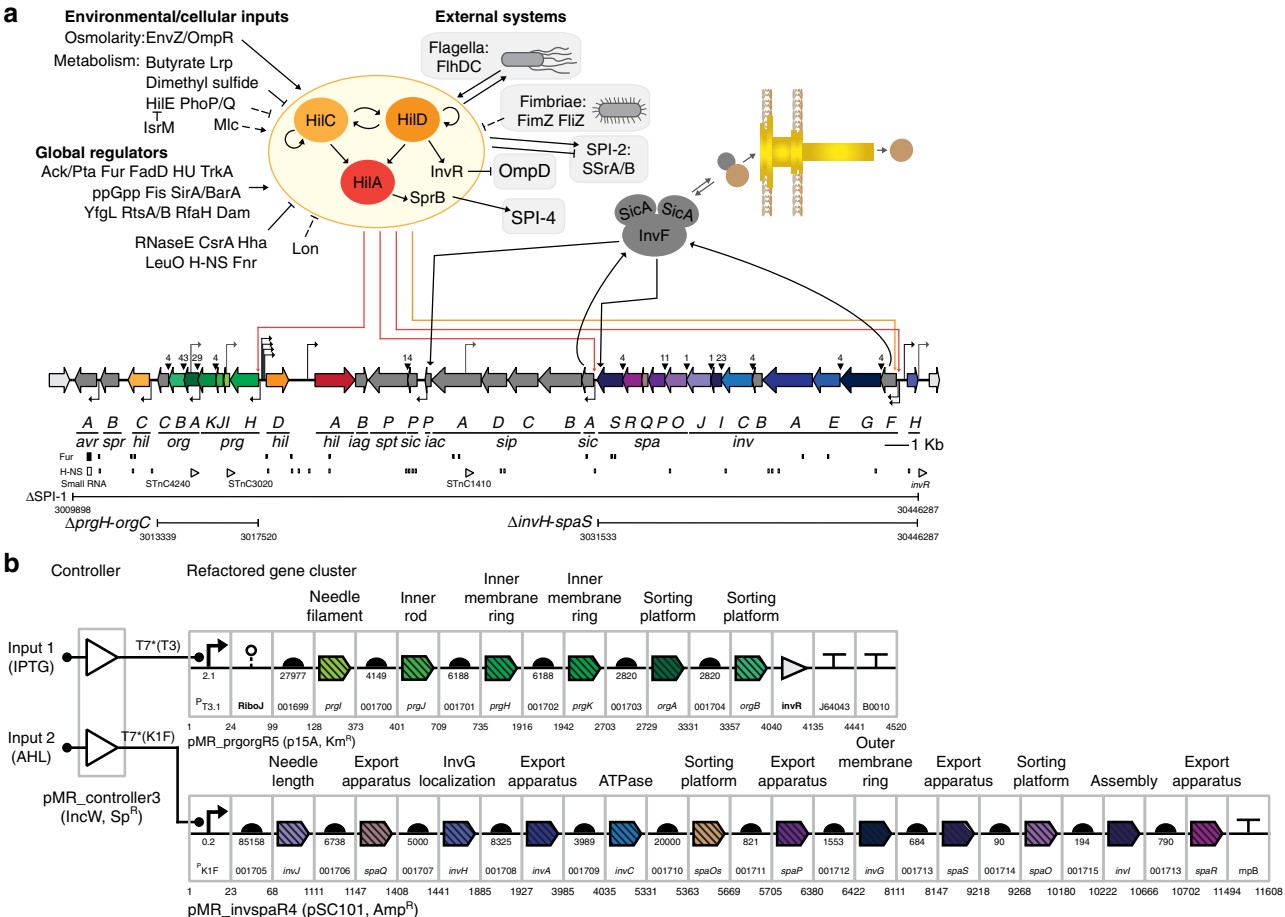

**Figure 1 | The native and refactored versions of SPI-1 are compared. (a)** SPI-1 is shown with the known internal and global regulatory network[24,58,67]. Transcription factors are shown in red/orange, effectors/chaperones are shown in grey, genes outside SPI-1 are white, and structural genes are coloured by operon occupancy. Transcriptional start sites for forward/reverse promoters are indicated by arrows above/below the DNA line[22,25,58,68]. Regions of translational coupling are marked with triangles and the number is the length of overlapping DNA. Predicted H-NS and Fur operator sites are shown[68]. Small RNAs are shown as empty triangles[25]. Knockouts described in this manuscript are shown along with the beginning and end nucleotide sequence, numbered to correspond to the LT2 genome sequence (NC_003197.1). **(b)** The complete refactored SPI-1 is shown, including the controller (*prg-org*R5*inv-spa*R4 and pMR_controller3). Each genetic part is surrounded by a grey box with the SBOLv (www.sbolstandard.org) symbol and SynBERC Registry (registry.synberc.org) part number at the bottom (each number is preceded by a SBa_). The number under the symbols shows the relative strengths of promoters and RBSs (Methods). The sequences of all genetic parts, including codon optimized genes, are provided in Supplementary Table 6. The plasmids containing the controller and two refactored operons are shown, with the full maps described in Supplementary Figs 17 and 18. The complete refactored system is shown, with intermediate constructs and strains described in Supplementary Table 2.

Note 1A and Supplementary Fig. 2). An initial construct was designed that radically changed the order of the genes (5′ to 3′): *invJ, spaQ, invA, invF, invC, spaP, invG, spaS, invE, spaO, invI, spaR*. The InvB chaperone was not included because it previously had been shown to be non-essential[42]. This reorganization and RBS selection disrupts 8 regions of translational coupling (Fig. 1a). The refactored *inv-spa* operon was synthesized as a single construct and placed under arabinose-inducible P_BAD control on a pSC101 plasmid. This was unable to complement secretion in a Δ*invF-spaS* knockout, even though secretion could be recovered when the native *invF-spaS* operon is expressed using the same inducible system and plasmid backbone (Fig. 2d).

To determine which gene(s) were causing the loss of function, a series of chimeric operons were tested that substituted each codon-optimized gene and RBS into the wild-type operon carried on the plasmid backbone (Fig. 2e and Supplementary Fig. 4). This narrowed the candidate to the recoded *spaO* gene, whose product oligomerizes to form the cytoplasmic C-ring[37]. Believing this to be due to the codon selection disrupting expression,

we redesigned the codon usage and synthesized a second gene, which also proved to be non-functional. From this, we hypothesized that the mutations could be disrupting an internal regulatory sequence. A series of chimeras were built that cross the synthetic and native sequences and a region in the centre of *spaO* (nucleotides 456–610) was found to be essential for function (Fig. 2f, Supplementary Note 1D). This region contains an alternative RBS and start codon (Fig. 2g) and when the start codon is mutated from *gtg* to *gtt*, this eliminates secretion (Fig. 2h, Supplementary Note 1D, Supplementary Table 4 and Supplementary Fig. 5). Using a western blot analysis confirmed by liquid chromatography–mass spectrometry (LC-MS) (see Methods, Supplementary Data 1), we discovered that there is a long (33 kDa) and short (11 kDa) protein that is tandem translated from *spaO* mRNA (Fig. 2i). After we discovered this, similar control was reported for SpaO homologues in the *Salmonella* SPI-2, *Shigella* and *Yersinia* T3SSs[43–45]. Our solution was to unpack the genetics[34] into independent long (*spaO*) and short (*spaOs*) genes that are physically separated in

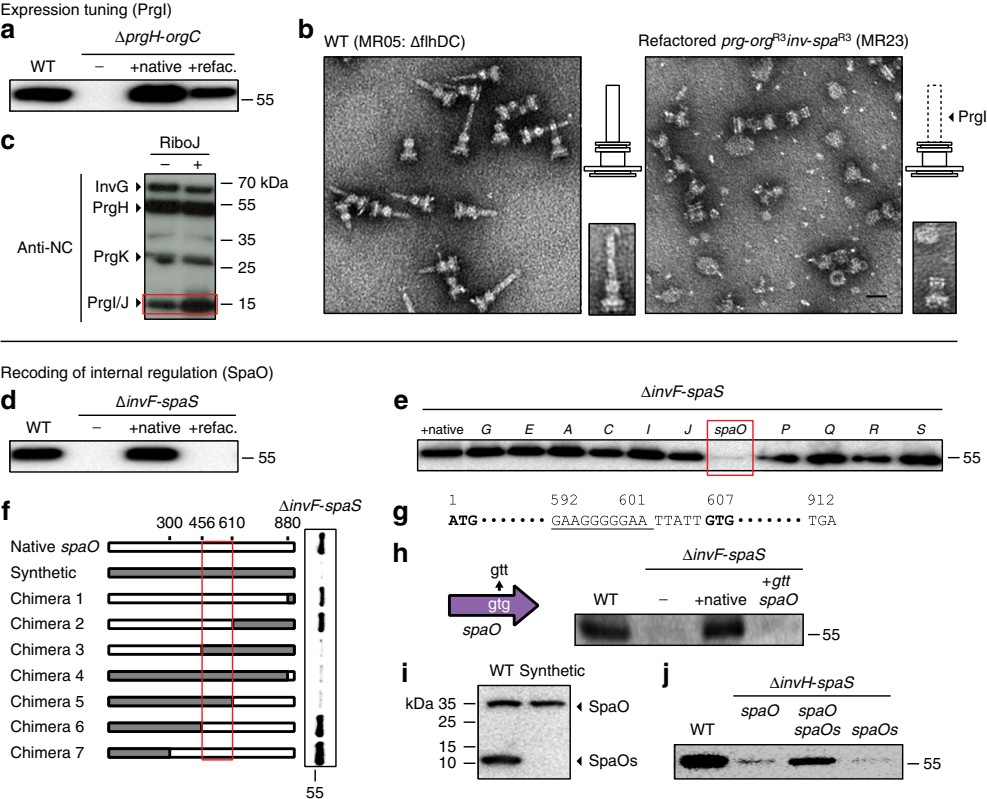

**Figure 2 | Intermediate steps in building and debugging the refactored SPI-1.** See Supplementary Table 2 and Supplementary Fig. 1 for the strain and variants of the refactored operons. See Methods for the description of the growth conditions and secretion assays. WT refers to *Salmonella typhimurium* DW01 and – is the knockout strain containing the reporter plasmid (Supplementary Fig. 16). The +native label refers to the knockout strain with the native operon under the control of the native promoter (*prg-org*) or $P_{BAD}$ (*inv-spa*, +0.1 mM arabinose) on a plasmid (Supplementary Fig. 21). (**a**) Secretion assays are shown for the Δ*prgH-orgC* strain complemented with *prg-org*R1 (+refac., +0.1 mM IPTG) on a plasmid. (**b**) Representative EM images are shown for negative stained needle complex isolated from a Δ*flhDC* strain containing intact SPI-1 versus the complementation of a double knockout with the refactored operons (*prg-org*R3 and *inv-spa*R3). The scale bar indicates 50 nm. (**c**) Needle complex components are visualized by western blot analysis, comparing the refactored operon lacking RiboJ (−, *prg-org*R3) or containing RiboJ (+, *prg-org*R4). (**d**) A secretion assay is shown for the first refactored *inv-spa*R1 operon. (**e**) A secretion assay is shown for chimeras made based on the first refactored operon where the native RBS and gene is swapped into each position (Supplementary Note 1C). (**f**) A set of *spaO* chimeras was made between the native sequence (white) and the codon optimized sequence (grey) within the refactored *inv-spa*R1 operon. The result of a secretion assay is shown. (**g**) The sequence of the native *spaO* gene is shown with the putative internal RBS underlined and start codon (GTG) in bold. (**h**) The internal start codon was mutated to GTT on the plasmid with native *inv-spa* operon. A secretion assay is shown for the complemented *inv-spa* (+native) compared to the single mutation (+gtt *spaO*). (**i**) Translated peptides from wild type or synthetic sequence of *spaO* was confirmed by western blot analysis. (**j**) Secretion assays are shown for variations of the refactored *inv-spa* operon that contains only full-length *spaO*, both *spaO* and the short *spaOs* (*inv-spa*R2), and only *spaOs*.

the refactored *inv-spa* operon (Fig. 1b, Supplementary Table 5, Supplementary Fig. 6). It is noteworthy that this also occurs in natural systems, where homologues to the long and short form appear as separate genes in the clusters encoding the *Salmonella* flagellum (*fliMN*) and the *Pseudomonas* T3SS[44,46].

A version of the refactored *inv-spa* operon containing the *spaO*, *spaOs* and *invH* genes was tested in the context of a Δ*invH-spaS* knockout. The separate encoding of the *spaO* genes is able to recover secretion and the lack of either gene causes an appreciable reduction in titre (Fig. 2j). At this point, the *invH* gene was also codon optimized and included in the refactored *inv-spa* construct (Supplementary Note 1E and Supplementary Fig. 7). In SPI-1, *invH* is transcribed in the opposite orientation as *invF-spaS*. While not essential for secretion, it has been shown to improve titre[13]. Incorporating it into the refactored *inv-spa* demonstrates that the separate transcription units can be encoded as a single operon.

**Design and construction of the complete refactored SPI-1.** A controller was then built to unify the regulation of the two refactored operons. A two-input two-output controller was

constructed for *Salmonella* (Fig. 3a)[38]. The two outputs are engineered RNAPs that bind to different promoter sequences and do not cross-react[38]. They are based on the T7 RNAP scaffold modified to be less toxic (T7*) and contain the DNA-binding loop from either T3 RNAP or K1F RNAP. The inputs to the controller are a tight IPTG-inducible system ($P_{tac-O2}$) and AHL-inducible system, which independently control T7*(T3) and T7*(K1F), respectively. Leaky expression was reduced by using weak RBSs to control the RNAPs, placing transcription units in opposite orientations, and using a low copy (IncW) plasmid (Supplementary Fig. 17). Each operon was then placed under the control of T7*(T3)- and T7*(K1F)-dependent promoters chosen to generate expression levels equivalent to the strength of the inducible systems used to test them individually (Supplementary Figs 3 and 8, Supplementary Notes 1B and 1F). The controller and refactored *prg-org* and *inv-spa* operons were first tested in a double knockout strain (Δ*prgH-orgC*Δ*invH-spaS*). This strain retains the remainder of SPI-1, including the region between *prgH* and *spaS* (containing the *hilD* and *hilA* regulators, 7 *sic-sip-spt* effectors and chaperones, *iagB*, and *iacP*) as well as a region

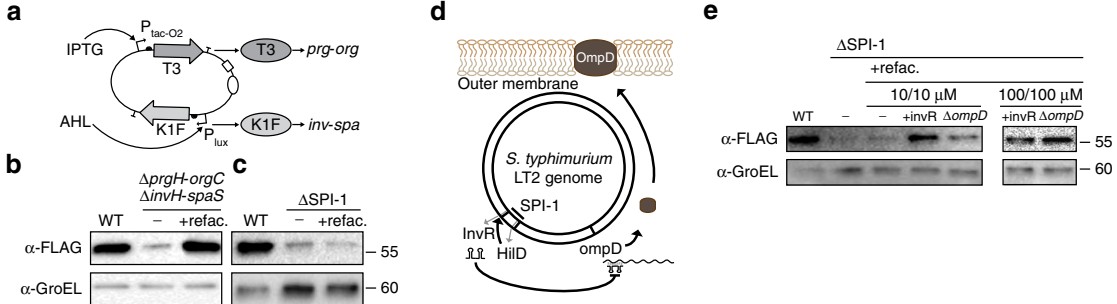

**Figure 3 | Completion of the refactored SPI-1.** (**a**) The dual controller is shown where T7*(T3) and T7*(K1F) are under IPTG- and AHL-inducible control, respectively (Supplementary Fig. 17). (**b**) A secretion assay is shown for a double knockout strain containing the two refactored operons (*prg-org*[R4] and *inv-spa*[R3]) with a controller ($+10\,\mu M$ IPTG/$+10\,\mu M$ AHL). The – lane is the absence of any complementation plasmid but with the inclusion of the reporter. (**c**) The secretion assay is shown for the duel (*prg-org*[R4] and *inv-spa*[R3]) and controller ($+10\,\mu M$ IPTG/$+10\,\mu M$ AHL) in $\Delta$SPI-1 strain. (**d**) An illustration of the repression of major *Salmonella* porin protein OmpD by small regulatory RNA, InvR upon activation by HilD encoded in SPI-1 (refs 1–3). (**e**) The secretion assay in a $\Delta$SPI-1 background complemented with the refactored operons (*prg-org*[R4] and *inv-spa*[R3]; lane 3) and controller ($+10\,\mu M$ IPTG/$+10\,\mu M$ AHL). The $+$invR lane corresponds to the presence of the small RNA (*prg-org*[R5] and *inv-spa*[R4], MR27). The $\Delta ompD$ strain contains the additional porin knockout (*prg-org*[R4] and *inv-spa*[R4], MR29).

upstream of *hilC* containing *sprB* and *avrA* (Fig. 1a). Upon the addition of both $10\,\mu M$ IPTG and $10\,\mu M$ AHL, the dual operon system recovers full secretion in the double knockout (Fig. 3b, Supplementary Note 1G).

Next, we tested the refactored operons (*prg-org* and *inv-spa*) and controller in a complete $\Delta$SPI-1 knockout. This knockout removes an additional 17.1 kb of DNA and the *sicA*, *sipBCDA*, *iacP*, *sicP*, *sptP*, *iagB*, *hilA*, *hilD*, *hilC*, *sprB* and *avrA* genes as compared to the $\Delta prgH$-*orgC*$\Delta invH$-*spaS* strain. In building the final operons, two additional genes hypothesized to be non-essential were removed and the preservation of secretion was confirmed (Supplementary Note 1I and Supplementary Fig. 11). The *invF* gene encodes a regulator that links the upregulation of effectors to the formation of functional needles[23]. The *invE* gene acts as a controller for translocation of SipBCD[3,47], which are no longer present in $\Delta$SPI-1.

Surprisingly, no secretion was detected in the $\Delta$SPI-1 strain (Fig. 3c). Further, there was an increase in GroEL, indicating a disruption in the integrity of the cell membrane. To determine the necessary missing component, a second knockout was constructed that reduces the amount of DNA deleted ($\Delta prgH$-*orgC*$\Delta invH$-*iacP*), limiting the possible issue to the *hilA*, *hilD* and *iagB* genes and intergenic DNA. The *iagB* gene was previously identified as non-essential[13] and its inclusion in the refactored cluster did not help (Supplementary Note 1H, Supplementary Fig. 10). This shifted the focus to the HilD and HilA regulators, which are known to regulate genes both within and outside of SPI-1 (refs 16,48,49). Notably, the *invR* gene is regulated by HilD and is within SPI-1, but is outside of the region removed to build the $\Delta$SPI-1 knockout (Fig. 1a). This encodes an abundant and stable sRNA regulator that represses the outer membrane protein OmpD (Fig. 3d)[50]. The *invR* gene was added to the 3′ end of the refactored *prg-org* operon (Fig. 1b) and this was found to be sufficient to recover secretion in the $\Delta$SPI-1 strain (Fig. 3e). The background GroEL also returned to wild-type levels. The mechanism was confirmed by making a $\Delta ompD$ knockout, which had a similar effect as expressing InvR (Fig. 3e). This is consistent with previous work showing that the misexpression of OmpD results in membrane disruption, which can be corrected by *invR* expression[50]. The complete functional refactored SPI-1 in the $\Delta$SPI-1 strain does not correspond to an increase in cell leakage or a decline in growth rate (Supplementary Fig. 14). In addition, the refactored SPI-1 was able to restore secretion in other *Salmonella* strains lacking SPI-1 (Supplementary Methods and Supplementary Fig. 15).

**Structure and function of the refactored T3SS.** The refactored cluster was then tested in media where native SPI-1 is strongly repressed. The standard means of inducing SPI-1 is in a high-osmolarity complex media (LB broth with 0.3 M NaCl). The refactored SPI-1 is active in this media, albeit at a lower titre ($6.7 \pm 1.4\,mg\,l^{-1}$) than wild-type levels ($8.0 \pm 0.9\,mg\,l^{-1}$) (Fig. 4a). The refactored system and SPI-1 produce similar titres in complex media at lower osmolarity (L-broth): $6.4 \pm 1.7\,mg\,l^{-1}$ from wild-type and $6.8 \pm 0.1\,mg\,l^{-1}$ from the the refactored system, respectively. SPI-1 is repressed in minimal media with low concentrations of phosphate or magnesium, which are representative of conditions in the macrophage[25,51,52]. As such, the native SPI-1 is strongly repressed in N salt minimal media with casamino acids as a carbon source (Fig. 4a). However, the refactored system can be induced with $10\,\mu M$ IPTG and AHL in this media and, leading to a titre of $3.4 \pm 0.5\,mg\,l^{-1}$.

The structure of the needle complex for the refactored *prg-org*[R5] and *inv-spa*[R4] operons in $\Delta$SPI-1 was then analysed by EM. Flagella interfere with the images, so an additional *flhDC* knockout was made to the $\Delta$SPI-1 strain and unmodified control strain. The needles were then purified, negative stained and compared using EM (see Methods). The lengths of the needles and the number of needles per cell are approximately the same for strains containing the native and refactored SPI-1 (Fig. 4b,c). The refactored SPI-1 (*prg-org*[R5] and *inv-spa*[R4] operons) produces needle complexes that are comparable in topology with the native cluster and are consistent with functional T3SSs observed in previous work[11,40,53]. RNA-seq of the refactored and WT clusters demonstrated similarity in the ratios of mRNA levels for each gene (Supplementary Methods and Supplementary Fig. 12). In addition, a western blot performed with purified needle complexes showed similar levels of protein expression, with the exception of lower PrgK and higher PrgJ in the refactored system (Supplementary Methods and Supplementary Fig. 13).

The controller enables the independent control of the refactored *prg-org* and *inv-spa* operons. We used this as a platform to quantify the sensitivity of secretion to changes in the relative expression levels and timing. These effects can be challenging to quantify using the native system because of the complexity of the regulation, including many redundant interactions and feedback control. The refactored cluster presents a clean system for changing the expression dynamics and measuring the impact on assembly or function. The refactored system was tested to determine if there is background activity due to leakage in

either inducible system (Fig. 4d). Secretion could only be detected in the presence of both inducers.

The assembly of the T3SS encoded in SPI-1 occurs in several steps. First, parts of the export apparatus (SpaPQRS) seed the inner membrane, then the inner and outer membrane rings are formed, and finally they are connected and the needle filament extends from the cell surface[3,13,14]. It has been postulated that the order of T3SS gene expression could reflect the order in which they are needed for assembly. For flagella, it has been observed that the temporal activation of the 'class 2' promoters (controlling the homologues to *prg* or *org* or *inv* or *spa*) reflects the order of assembly[54,55]. A similar ordering of promoter activities has been observed for T3SSs involved in pathogenesis[56]. For SPI-1, it has been observed that the needle complex cannot form if the SpaPQRS genes are expressed 5 h after the genes that form the base[14]. Using the refactored SPI-1, we changed the relative times that the *prg-org* and *inv-spa* operons are induced (Fig. 4e). One inducer is always added at $t = 0$ h and the second is added

after a delay and secretion is quantified at $t = 6$ h (see Methods). The FLAG tagged SptP secretion reporter is expressed from a constitutive promoter; thus, its total expression over the course of the experiment is constant. For differences in timing less than 3 h, there is no difference in secretion titre: the two operons can be expressed simultaneously or in any order with no effect. After 3 h, there is a significant difference in expressing *inv-spa* before *prg-org*, as compared to the opposite order. However, we find that the optimal amount of secretion occurs when both inducers are added simultaneously, indicating that delays on the timescale of transcription and translation do not impact the formation of functional needle complexes.

## Discussion

SPI-1 is regulated by numerous factors and the list has been growing for years[22]. It has been increasingly difficult to untangle the regulation and determine the importance and role of the individual elements. Here, we have taken a minimalist approach and have re-built SPI-1 from the bottom-up in modular form with completely synthetic parts and regulatory control. This has shown that none of the regulatory motifs or architectural features of the native cluster are required to generate functional needles. Even when essential features were encountered, we were able to recode them in ways that differ from the organization of SPI-1 (e.g., unpacking of *spaO*, constitutive control of *invR*, transcript stability for *prgI*). This demonstrates that there are many equivalent ways to encode this function and, indeed, related T3SS clusters across species shows significant genetic variation[57]. Thus, the majority of SPI-1 regulation appears to coordinate it with other virulence mechanisms that require precise timing in the host[58]. Two additional roles for regulation may also provide benefit to the native cluster. First, some regulation, such as the temporal order of gene expression and translational coupling, may not be required for function and rather exists to efficiently utilize cellular resources[54,59]. Second, non-essential genes and

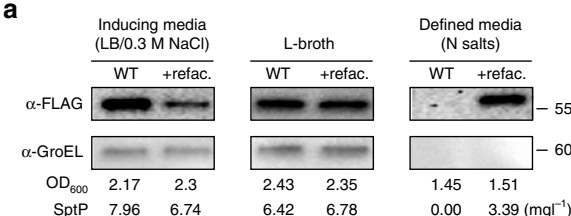

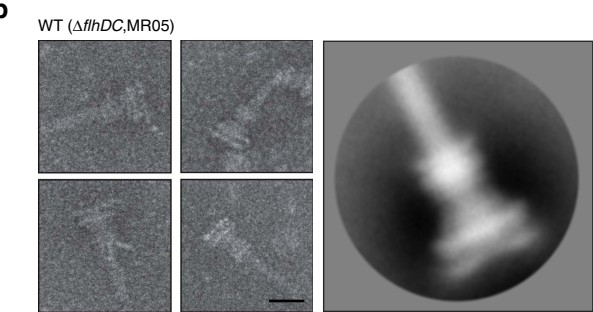

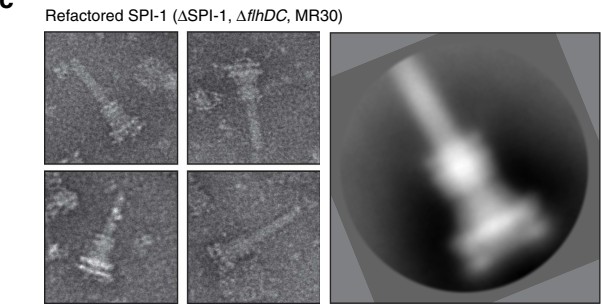

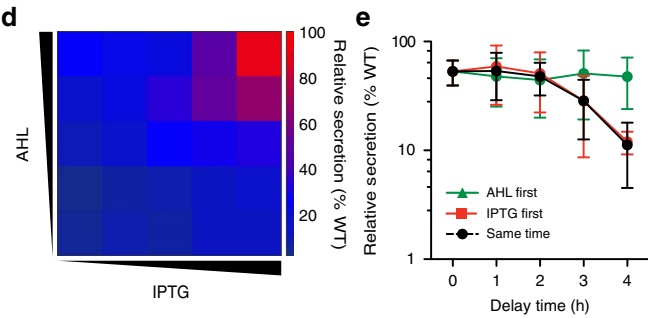

**Figure 4 | Structural and functional analysis of the refactored T3SS.** Data are shown for the complete refactored system (Fig. 1b, *prg-org*[R5] and *inv-spa*[R4]) in the ΔSPI-1 strain background (MR27). (**a**) Cultures were grown in different media (Methods). The refactored system was induced with 10 μM IPTG and 10 μM AHL. The quantification of secreted proteins is described in Methods. Bacterial growth as $OD_{600}$ and secreted protein titre are noted at the time of sampling. The growth curves of each culture are shown in Supplementary Fig. 14. (**b**) Representative images of T3SS needle complexes isolated from *Salmonella typhimurium* LT2 DW01 (WT) with the flagella knockout are shown (MR05). The image on the right represents a class average from 129 individual particles (see Methods). (**c**) Same as part **b**, with the refactored SPI-1 (*prg-org*[R5] and *inv-spa*[R4]) in a Δ*flhDC*ΔSPI-1 background (MR30). The image on the right represents a class average from 158 individual particles. The scale bar indicates 50 nm. The images of individual particles are shown in Supplementary Fig. 24. (**d**) Relative secretion is calculated at the band intensity of a secretion assay for the refactored SPI-1 divided by the intensity obtained from the wild-type (Methods). A secretion assay was performed 6 h after induction. The following concentrations of inducer were added to the culture: IPTG (0, 0.1, 1, 10, 100 μM) and AHL (0, 0.1, 1, 10, 100 μM). Data represent averages of three replicates performed on different days. (**e**) The impact of delays between the expression of *prg-org*[R5] and *inv-spa*[R4] are shown (MR27). For the green and red lines, one inducer is added at $t = 0$ and then the secretion titre is measured at $t = 6$ h. The relative secretion is calculated as described in part **d**. For the black line, both inducers are added at the indicated time point and the secretion measurement is made at $t = 6$ h. Either inducer is added to a concentration of 10 μM. Data shown are the average of three experiments performed on different days and the error bars represent the s.d.

regulation may be important to increase the range of cell states and environmental conditions where the T3SS can function. Indeed, InvR has been speculated to aid the establishment of SPI-1 post-transfer by clearing porins from the new host[50]. A refactored system offers a platform to test hypotheses regarding the role of regulation or genetic organization. Removing native regulation and modularizing genetics creates a clean background where changes can be made and interpreted in isolation and this enables the comparison of large-scale changes or alternative designs.

## Methods

**Strains and media.** The *Salmonella enterica* serovar Typhimurium LT2 DW01 strain and isogenic mutant strains used for all experiments are listed in Supplementary Tables 2 and 3. The DW01 lab strain was sequenced and is provided in Supplementary Note 1A and Supplementary Table 1. Each mutant strain was generated by the λ-red-mediated gene replacement system and oligonucleotides used for the generation of mutant strains are listed in Supplementary Table 3 (ref. 60). Genomic mutations were moved into DW01 by P22HT *int*-mediated transduction, including the double knockout strains and the nonflagellated mutant ($\Delta flhDC$)[7]. In-frame deletions were generated by recombining the two FRT sites with the Flp recombinases encoded in the pCP20 plasmid. Bacterial cells were grown in L-broth (LB Broth base; Invitrogen #12780-052) or SPI-1-inducing media (IM; LB containing 0.3 M NaCl) (LB Miller Medium, Difco #244620; sodium chloride, amresco #X190) with antibiotics if necessary: chloramphenicol (34 µg ml$^{-1}$; USB #23600), ampicillin (50 µg ml$^{-1}$; sodium ampicillin, Gold Biotechnology #A-301-25), kanamycin (50 µg ml$^{-1}$; kanamycin sulfate, Gold Biotechnology #K-120-10) and spectinomycin (50 µg ml$^{-1}$; spectinomycin dihydrochloride, Gold Biotechnology #S-140-5). For defined media samples, cells were grown in N salt minimal media (5 mM KCl (Sigma-Aldrich #P5405), 7.5 mM (NH$_4$)$_2$SO$_4$ (EMD Millipore #AX1385), 0.5 mM K$_2$SO$_4$ (Fisher Scientific #P304-500), 1 mM KH$_2$PO$_4$ (Sigma-Aldrich #795488), 38 mM glycerol (BDH), 0.1% casamino acids (Difco #223050) supplemented with 100 mM Tris-HCl, pH 7.6) with 10 mM or 8 µM MgCl$_2$ (USB #1864). Unless otherwise noted, the refactored clusters were induced by 100 µM L-(+)-arabinose (Sigma #A3256), N-(β-ketocaproyl-L-homoserine lactone (AHL; Cayman Chemical #10011207) and/or 100 µM isopropyl-β-D-1-thiogalactopyranoside (IPTG; Gold Biotechnology #I2481C) supplemented into IM.

**Growth Curves.** The bacterial growth was measured from the strain expressing wild type or the refactored T3SS in inducing media (IM) or rich media without high osmolarity (L-broth) or defined media (N salt) (Figure 4a). Each bacterial culture was taken every hour after shifting into each media for the measurement of optical density and the refactored system was induced by addition of IPTG and AHL (10 µM, respectively).

**ORF design and gene synthesis.** The sequence of each gene was initially designed by DNA2.0 (Mountain View, CA, USA) to simultaneously maximize the Hamming distance from the native sequence and optimize the codon usage for *Salmonella*. All *cis*-elements including restriction sites, RBS and transcriptional terminators, were eliminated by the DNA2.0 algorithm or via software that we ran (Prokaryotic Promoter Prediction Server for σ$^{70}$-promoters[61], PromScan algorithm for σ$^{54}$-promoters[62] and TransTermHP software for terminators[63]). DNA was synthesized, typically as complete operons, by DNA2.0.

**Synthetic RBS design.** The first RBS sequences were obtained from The Registry of Standard Biological Parts (http://parts.igem.org): BBa_B0030, BBa_J61101, BBa_J61130. These RBSs were cloned into *Xba*I/*Sac*I site between constitutive promoter (BBa_J23100) and red fluorescence protein, mRFP, on a ColE1 plasmid (Supplementary Fig. 20), which was modified from pPROtet.E133 (Cm$^R$; Clonetech). Then RBS library was constructed by random mutagenesis using error-prone PCR. Strength of each synthetic RBS from library was measured using mRFP as artificial unit (a.u.) with flow cytometry and screened to adjust the strength to native RBS (Supplementary Note 1A). For design of RBS for *spaOs* and *invH*, RBS Calculator online tool version (https://salis.psu.edu/software) was used: the RBS Calculator Design mode version 1.1, with organism option as *Salmonella enterica* subsp. *enterica* serovar Typhimurium str. LT2 (ACCTCCTTA).

**Characterization of genetic parts.** To measure RBS strength, bacterial strains were grown in L-broth with antibiotics at 37 °C overnight and then freshly inoculated at 200-fold dilution into fresh L-broth with antibiotics in a 96-well plate (Nunc #249952) in an ELMI Digital Thermos Microplates shaker incubator (Elmi Ltd, Riga, Latvia) at 1,000 r.p.m. and 37 °C. After 2 h of incubation, cells were diluted 1:10 into IM and cultured under the same shaking incubator parameters for 6 h. An aliquot (5 µl) of cell culture was added into 195 µl of phosphate-buffered saline with 0.5 mg ml$^{-1}$ kanamycin to arrest cell growth and then analysed by flow

cytometry. For measurement of promoter strength on a refactored plasmid, each refactored cluster was replaced with fluorescence protein: *prg-org* cluster with sfGFP on p15A plasmid and *inv-spa* with mRFP on pSC101 plasmid (Supplementary Fig. 19). Both plasmids were then introduced into DW01 strain harbouring a controller plasmid pMR_controller3. Upon induction by addition of 10 µM IPTG and AHL, promoter strength was reported as relative expression units, which were calculated by normalizing the sfGFP or mRFP output values by that measured from a reference standard. White-cell fluorescence measured from a strain lacking fluorescence protein was subtracted from measured fluorescence before normalization. DW01 strain harbouring P$_{J23101}$-sfGFP or P$_{J23101}$-mRFP was used as reference standard, respectively, which has a constitutive promoter (BBa_J23101) followed by a 5′ UTR (BBa_B0032) and sfGFP or mRFP (Supplementary Fig. 19).

**Flow cytometry analysis.** Data were obtained using an LSRII flow cytometer (BD Biosciences). All the data, consisting of up to 100,000 cells, were gated by forward and side scatter. The geometric mean values of fluorescence were calculated with FlowJo (TreeStar Inc.). White-cell fluorescence measured from a strain lacking fluorescence protein was subtracted.

**Secretion assay.** The overnight cultures in L-broth were diluted to an OD$_{600}$ of 0.02 into 6 ml fresh L-broth in a 14 ml polypropylene round-bottom tube (Falcon #352059) and then grown for 2 h at 250 r.p.m. The cultures were diluted 1:10 into 50 ml IM in a 250 ml erlenmeyer flask, unless otherwise stated, containing antibiotics and inducers if necessary and grown at 37 °C for 6 h at 160 r.p.m. until the OD reached 1.8–2.5. Supernatants were prepared by filtration through the 0.2 µm filter unit (Nalgene) after centrifugation at 15,000 g for 3 min. The secreted proteins in supernatants were analysed by western blots against FLAG-tagged SptP, which was encoded on a reporter plasmid (pMR_reporter; Supplementary Fig. 16) or representative cytoplasmic protein, GroEL as a cell lysis control. For cultures grown in defined media, this procedure was followed except cells are grown in N salt minimal media (+10 mM MgCl$_2$) overnight and diluted to OD$_{600}$ of 0.2 in 50 ml of N salt minimal media (+8 µM MgCl$_2$) containing antibiotics and inducers and grown at 37 °C for 8 h at 160 r.p.m. until the OD$_{600}$ reached 1.0–1.4.

**Western blot analysis.** The supernatant was prepared in SDS sample buffer containing 0.2% β-mercaptoethanol and then run on 10% SDS-PAGE gel (Lonza #58502). The gels were transferred onto PVDF membrane (Biorad #162-0177) and then blocked for 1 h in 5% skim milk (w/v of TBST) at room temperature. The anti-Flag antibody (Sigma #F3165) or the anti-GroEL antibody (Sigma #G6532) was used as a primary antibody. Each antibody was added 1:2000 (v/v) into 2.5% skim milk (w/v of TBST) and then allowed to bind for 1 h at room temperature. The membranes were washed with TBST three times after each antibody-binding step. The HRP-conjugated anti-mouse antibody (Sigma #A8924) was used as a secondary antibody for the detection of Flag-tagged proteins, and for the detection of GroEL from supernatant as lysis control, the HRP-conjugated anti-rabbit antibody (Sigma #6154) was used as a secondary antibody. The secondary antibody (1:4,000) was incubated for 1 h at room temperature. After washing the membrane, the signal was developed by chemiluminescence for HRP (Pierce #32106) and detected using the Biorad chemidoc MP imaging system (#170-8280). Gel densitometry was done using ImageJ 1.41 (NIH). The relative secretion presented in Fig. 4d,e was calculated as a percent intensity by dividing the intensity of the band by that measured for wild type and multiplying by 100. Uncropped images of blots are shown in Supplementary Fig. 23.

**Quantitative western blot analysis.** The titre of secreted FLAG-tagged SptP was determined by quantitative western blot using the Cy-5 conjugated secondary antibody (GE Healthcare Life Sciences #PA45009). The supernatant samples were prepared and run on the SDS-PAGE gel as described in western blot analysis. A standard curve was generated by running of carboxy-terminal FLAG-BAP Fusion Protein (Sigma #P7457) on the gel together with each sample. The gel was transferred onto Hybond low fluorescence PVDF membrane (AmershamBiosciences Cat#: RPN303LFP). After primary antibody binding followed by the washing step, the anti-mouse IgG-Cy-5 antibody was used as the secondary antibody. The secondary antibody was added 1:2,000 (v/v) and incubated for 1.5 h at room temperature. After washing the membrane, the membranes were dried and then the signal was imaged by the Biorad chemidoc MP imaging system. ImageJ was used to analyse gel densitometry. Protein quantities were determined using the standard curve and linear fit.

**Immunoprecipitation and LC-MS.** Native or synthetic coding sequence of *spaO* was amplified by PCR with 3′ primer containing 3x Flag sequences then cloned into pEXT-21 (ref. 64) by ligation of *Sac*I–*Kpn*I fragment (Supplementary Fig. 22). DW01 containing each plasmid was cultured under the same condition for secretion assay and the expression of native or synthetic *spaO* was induced by the addition of 1 mM IPTG into IM. Ten millilitres of each cell was harvested by centrifugation at 6,000 g. Cell pellets were washed with lysis buffer (100 mM NaCl,

25 mM Tris-HCl pH 8.0) and then resuspended in lysis solution containing Bugbuster (Novagen #70921) in buffer with 250 Units of Benzonase (Sigma #E1014). Unbroken cells were eliminated by centrifugation at 16,000g, 4 °C for 10 min. The cell lysates were analysed by western blot against the anti-Flag antibody and also applied to immunoprecipitation against anti-Flag M2 affinity gel (Sigma #A2220). As recommended by the manufacturer, the affinity resin was equilibrated by washing with Tris-buffered saline (TBS; 50 mM Tris-HCl pH 7.4, 150 mM NaCl) twice and then resuspended in TBS. Cell lysates were mixed with equilibrated resin on a rocker at 4 °C for 2 h. Unbound proteins were eliminated by centrifugation at 8,000g for 30 s followed by washing with TBS. Supernatant was discarded using narrow-end pipette tip. For elution of bound proteins, 2 × SDS loading buffer without DTT was added into the mixture and then samples were boiled for 3 min. Undissolved resin was eliminated by centrifugation at 8,000g for 30 s. The sample was run on 4–20% gradient SDS-PAGE gel (Lonza #59551) and then the gel was stained with coomassie blue (Biorad #1610786). The band for small peptide (approximately 11–12 kDa), which was only from native spaO gene, was cut from the gel and then analysed (Biopolymers and Proteomics Core Facility at Koch institute in MIT). The excised band was subjected to in-gel disulfide reduction, cysteine alkylation with iodoacetamide and digestion with trypsin. Separation of proteolytic peptides was carried out by nanoflow high-performance liquid chromatography gradient elution using a capillary reversed phase C18 column. Peptides were analysed by LC-MS. Peptide molecular weight measurements and fragments ion mass spectra were obtained with a QSTAR Elite quadrupole time-of-flight mass spectrometer (Supplementary Data 1).

**Needle complex purification and EM.** The experiments were conducted with nonflagellated strain derived from DW01 strain (MR05: ΔflhDC). The needle complexes were isolated from ΔSPI-1 with prg-org[R5] and inv-spa[R4] (Refactored SPI-1) and MR05 (WT: ΔflhDC). The growth conditions were same as for secretion assays as well as for the isolation of needle complex[53] with several modifications. Briefly, cells were harvested after 6 h culture in IM and then harvested cells were resuspended in 150 mM Tris-HCl (pH 8.0), 0.5 M sucrose (Sigma-Aldrich #84097), 0.5 mg ml$^{-1}$ hen egg lysozyme (Fluka #62971), 5 mM EDTA (AppliChem #A1103) and kept on ice for 45 min and then incubated at 37 °C for 15 min. Cells were lysed with 0.3% lauryldimethylamine oxide (LDAO; Sigma-Aldrich #40234) before adding 10 mM MgCl$_2$ and 500 mM NaCl to the lysate. After removal of cell debris by low-speed centrifugation, needle complexes were pelleted by high-speed centrifugation (Beckman, 45Ti rotor, 140,000g, 4 h, 12 °C). The pellet was resuspended in 0.5% LDAO in 10 mM Tris-HCl (pH 8.0), 0.5 M NaCl, 5 mM EDTA and adjusted to a final concentration of 30% w/v of CsCl (amresco #0415). Two millilitres of resuspended samples were centrifuged for 12 h at 210,000g in a Beckman TLS-55 rotor. Half-millilitre aliquots were combined with 2.5 ml CsCl-free buffer and samples were pelleted in a TLA-110 rotor at 430,000g for 30 min. The complexes were resuspended in 0.1 ml 0.2% LDAO, 10 mM Tris-HCl (pH 8.0), 0.5 M NaCl and stored at 4 °C until image analysis. Western blot analysis of needle complex preparations was carried out with anti-needle complex antibody (NC). Thin carbon films were obtained by evaporating with an Edwards E306 machine onto mica sheet. One day before usage these films were floated onto 400 mesh hexagonal Cu/Pd-grids (3.05 mm; Agar Scientific #AGG2440PD). Samples was applied to glow-discharged grids and negatively stained with 2% phosphotungstic acid (Sigma-Aldrich #P4006) and then observed at × 56,000 nominal magnification with a Morgagni transmission electron microscope (FEI) operated at 80 kV. In addition, micrographs for image data processing were recorded on at a Tecnai T20 transmission electron microscope operated at 200 kV on a 4 K-CCD at nominal magnification of 50,000 resulting in 2.18 Angstrom/pixel or a Polara FEI transmission electron microscope set at 300 keV and a nominal magnification of 59,000 (2.09 Angstroem/pixel) microscope. Image data processing, including determination of the contrast transfer function and 2D-classification, was performed using the software CTFFIND3 and RELION, respectively[65,66]. In addition, isolated needle complex was also analysed by western blot analysis against anti-needle complex antibody.

**Data availability.** The authors declare that all data supporting this study are available within the article and its Supplementary Information file or are available from the corresponding author upon request. All RNA-seq data are available in Sequence Read Archive (Accession codes: SRR5176473, SRR5176472).

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

## Acknowledgements

This work is supported by NIHR01-AI067699 and NIHP50-GM098792; National Science Foundation (SA5284-11210 grant to the Synthetic Biology Research Center); Pew Foundation. D.B.G. was supported by the US Defense Advanced Research Projects Agency (DARPA) Living Foundries grant HR0011-13-1-0001. T.C.M was generously supported through the Behörde für Wissenschaft, Forschung und Gleichstellung of the city of Hamburg. Electron Micrograph samples were recorded at the EM facility of the Vienna Biocenter Core Facilities GmbH (VBCF), Austria.

## Author contribution

M.S., D.J.S., E.A.M. and C.A.V. designed the experiments. M.S. and C.A.V. wrote the manuscript. M.S., D.J.S., E.A.M. and A.C.T. performed the experiments and analysed the data. L.C., J.M., J.F.R. and T.C.M. performed needle complex isolation and analysis. D.B.G. performed the genome sequencing and genotype analysis.

## Additional informations

**Competing interests:** The authors declare no competing financial interests.

