## [Peer Review File · Nature Communications]

Reviewers' Comments:

Reviewer #1 (Remarks to the Author)

The authors constructed a controllable, modular and simplified version of the type III secretion network from Salmonella (SPI-1). This refactoring approach led to the discovery of the critical roles of specific factors including SpaO and InvR. The refactored T3SS is an important achievement that could be used for biotechnology applications. In addition, future work could dissect this simplified network to explore the mapping between the expression of each component and the secretion titer to incrementally improve the performance.

However, I think this work is better suited for a specialized journal as opposed to a journal with a broad readership. The design process is not sufficiently novel compared to previous works that used similar methods to refactor the Nitrogen fixation gene cluster and T7 bacteriophage genome (PMID: 16729053, 22509035, 25419741). It is not clear whether methods used and insights gleaned from this work could be applied to improve the design process of different gene clusters and networks. Beyond the discoveries of the key functional role of specific (SpaO and InvR), the authors do not provide novel biological insights into the T3SS network organizational principles. For example, a quantitative comparison of the expression of each component in the WT and the refactored network could be informative. Finally, the secretion titer of the final refactored pathway is lower than the WT system based on Figure 3d,e.

An interesting question is how to design the refactored network to exhibit a higher secretion titer than WT. The timing of gene expression is likely an important parameter and could be explored further. For example, the authors could design controllers to regulate specific submodules of the network to mirror the dynamic regulation of the WT system.

Below are some specific comments about the manuscript:

- There are several spelling errors in the abstract and introduction sections
- "Stringent" in the Introduction section could be misinterpreted to mean the "stringent response"
- The introduction section is missing an overview paragraph outlining the major findings
 - The authors claim the network to be "minimal" but they do not provide evidence for this claim. Does minimal refer to total number of genes or pathway length in bp?
 - It would be helpful if the authors could provide quantitative information about the specific number of constructs tested and increase in titer as opposed to "many" or "higher".
 - It is surprising that the expression of the T3SS system does not incur a growth or lag phase defect. For example, the OD600 value is similar for the WT and +Refac in Figure 3a even though T3SS is active in +Refac. It would be useful if the authors showed growth curves of cells expressing the WT and refactored T3SS networks to support the conclusions about fitness costs in Figure 3a.

Reviewer #3 (Remarks to the Author)

The authors report a synthetic biology study in which they manage to de-couple all elements of natural control of order and level of expression of type three secretion system components but still assemble the nano-machine across the bacterial membranes. The machine so assembled is demonstrated to be active in secretion.

This reviewer is not a frequent reader of synthetic biology papers and found the language (e.g. refactored) somewhat obfuscating at times. Otherwise the paper is well written and clearly describes the results.

The approach described here holds much promise for dissection of the control of assembly and activity of these systems and as such is a significant advance.

Major Comment

The data presented in Figure 3 require a small amount of extension as they do not allow the level of correct assembly of the system to be assessed - the data presented in panel C certainly suggest that some grossly correctly assembled machines are generated (I assume the the panel on the right shows a single class average rather than a single particle?), but the higher level of background in the raw micrographs shown on the left perhaps suggests yields were lower? At best these images are only proof that there is no gross, morphological, flaw in the refactored system. Whilst the data presented in D and E suggest close to wild-type levels of secretion can be achieved, the lack of quantification of the levels of protein expression do not allow the reader to appreciate if the level of correct assembly differs in the WT and refactored systems. Such data should not be problematic to generate and would significantly strengthen the manuscript. With this addition I am happy recommend publication of this significant piece of work.

Minor comments.

-the citations should be completed by noting that the translation of short and long forms of SpaO homologs has also been reported by McDowell et al, Mol Micro 2016

Reviewer #1:

1. *I think this work is better suited for a specialized journal as opposed to a journal with a broad readership. The design process is not sufficiently novel compared to previous works that used similar methods to refactor the Nitrogen fixation gene cluster and T7 bacteriophage genome (PMID: 16729053, 22509035, 25419741). It is not clear whether methods used and insights gleaned from this work could be applied to improve the design process of different gene clusters and networks. Beyond the discoveries of the key functional role of specific (SpaO and InvR), the authors not provide novel biological insights into the T3SS network organizational principles. For example, a quantitative comparison of the expression of each component in the WT and the refactored network could be informative. Finally, the secretion titer of the final refactored pathway is lower than the WT system based on Figure 3d,e.*

An interesting question is how to design the refactored network to exhibit a higher secretion titer than WT. The timing of gene expression is likely an important parameter and could be explored further. For example, the authors could design controllers to regulate specific submodules of the network to mirror the dynamic regulation of the WT system.

New experiments have been included to elucidate biological insights into the T3SS network organizational principles. This includes RNA-seq and protein expression analyses performed on the WT and refactored systems that enables us to determine the required ratios for gene expression (page 15-16).

We have also quantified the secretion titers and this information is provided. Note that Figure 3a shows the right data to compare the WT and refactored systems. The process of refactoring eliminates regulation, thus the engineered system performs consistently across medias. For example, the activity in N salts is many-fold higher than WT (which is undetectable).

The nitrogen fixation (*nif*) system published previously from our lab was different because the regulation of the natural *nif* gene cluster is simple – in essence, a single transcription factor – and the number of signals integrated is small and turns on the whole pathway under the right conditions. In contrast, SPI-1 is totally the opposite situation with an incredible number of regulatory factors being integrated at all levels of the pathway and complex gene expression dynamics are implemented (see Figure 1). It was not at all clear what aspects of the regulation had to be preserved for function and, in fact, it is remarkable how much can be removed and the core remaining set constitutes new biological insights. The T7 phage genome refactoring done by a different lab is a completely different approach (despite using the term “refactoring”) and is not comparable.

2. *There are a several spelling errors in the abstract and introduction sections*

These have been corrected.

3. *“Stringent” in the Introduction section could be misinterpreted to mean the “stringent response”*

This has been changed to “strict.”

4. *The introduction section is missing an overview paragraph outlining the major findings*

The introduction has been adjusted to make this clearer.

5. *The authors claim the network to be “minimal” but they do not provide evidence for this claim. Does minimal refer to total number of genes or pathway length in bp?*

Minimal refers to the elimination of regulation (and elimination of regulatory and unnecessary genes).

6. *It would be helpful if the authors could provide quantitative information about the specific number of constructs tested and increase in titer as opposed to “many” or “higher”.*

We now present the quantitative titers. Multiple students performed the project in pieces and the number of constructs would not be informative or possible for us to ascertain. We have attempted to capture the complexities of this process in Section II of the SI, which summarizes the design process. The project did not involve large libraries of alternative part substitutions that we have presented as part of other pathway optimization projects (in part because the secretion assay and associated controls are low-throughput). Only a relatively few constructs of the complete pathway were built.

7. *It is surprising that the expression of the T3SS system does not incur a growth or lag phase defect. For example, the OD600 value is similar for the WT and +Refac in Figure 3a even though T3SS is active in +Refac. It would be useful if the authors showed growth curves of cells expressing the WT and refactored T3SS networks to support the conclusions about fitness costs in Figure 3a.*

New growth curves are provided in supplementary section VI.

Reviewer #3:

1. *This reviewer is not a frequent reader of synthetic biology papers and found the language (e.g. re-factored) somewhat obfuscating at times.*

We have carefully edited the text to either remove or better define jargon.

2. *The data presented in Figure 3 require a small amount of extension as they do not allow the level of correct assembly of the system to be assessed - the data presented in panel C certainly suggest that some grossly correctly assembled machines are generated (I assume the panel on the right shows a single class average rather than a single particle?), but the higher level of background in the raw micrographs shown on the left perhaps suggests yields were lower? At best these images are only proof that there is no gross, morphological, flaw in the refactored system.*

The images on right panel of Figure 3b and 3c are showing a single class average of classified particles from each strain and actual particle numbers were included in the caption of Figure 3.

3. *Whilst the data presented in D and E suggest close to wild-type levels of secretion can be achieved, the lack of quantification of the levels of protein expression do not allow the reader to appreciate if the level of correct assembly differs in the WT and refactored systems. Such data should not be problematic to generate and would significantly strengthen the manuscript.*

The expression levels of components were examined by RNA-seq and western blot analysis for the WT and refactored systems. These data are provided in the Supplementary Information.

4. *The citations should be completed by noting that the translation of short and long forms of SpaO homologs has also been reported by McDowell et al, Mol Micro 2016.*

The paper has been added into the reference list.

The authors did not address the two previous comments below:

- Figure 2 is dense with many results and the authors could consider breaking the panels into two separate figures and including some schematics to highlight the main findings.
- The authors could consider including a schematic showing the refactoring design process

Regarding the RNA-seq scatter plot (Figure S12), it is not clear what the data points represent. According to the Figure legend, the data points denote genes in the T3SS pathway. However, the total number of data points significantly exceeds the number of listed genes. Further, the authors should quantify the difference in expression for each gene and evaluate the correlation and statistical significance of the data. The scatter plot shows that the WT and refactored pathway exhibits differential transcript abundance profiles.

Other comments:

- The organization of the manuscript could be improved. For example, the authors could consider adding sub-sections in their results section.
- The introduction could be expanded to contain relevant background information on refactoring and key network interactions in the T3SS pathway.
- It would be helpful if the authors could compare their design approach to previous refactoring efforts such as the Nitrogen fixation pathway in the discussion section. Does this study provide new insights into refactoring of complex networks?

Reviewer :

1. *The authors did not address the two previous comments below:*
 - a. *Figure 2 is dense with many results and the authors could consider breaking the panels into two separate figures and including some schematics to highlight the main findings.*

Figure 2 is now separated into two figures (Figure 2 and 3).

- b. *The authors could consider including a schematic showing the refactoring design process*

A schematic diagram for refactoring design process is described in detail and also referred to our previous paper detailing core principle for the process. We do not believe that reproducing this information as a schematic is critical.

2. *Regarding the RNA-seq scatter plot (Figure S12), it is not clear what the data points represent. According to the Figure legend, the data points denote genes in the T3SS pathway. However, the total number of data points significantly exceeds the number of listed genes. Further, the authors should quantify the difference in expression for each gene and evaluate the correlation and statistical significance of the data. The scatter plot shows that the WT and refactored pathway exhibits differential transcript abundance profiles*

The figure legend has been edited for clarity. We have quantified the difference in expression of each gene and present the correlation and significance of the data (Supplementary Figure 12).

3. *The organization of the manuscript could be improved. For example, the authors could consider adding sub-sections in their results section.*

We have divided the text into subsections and have added subsection titles.

4. *The introduction could be expanded to contain relevant background information on refactoring and key network interactions in the T3SS pathway.*

We have added citations and edited the text for clarity.

5. *It would be helpful if the authors could compare their design approach to previous refactoring efforts such as the Nitrogen fixation pathway in the discussion section. Does this study provide new insights into refactoring of complex networks?*

The major difference between the two is in dealing with the regulation. Nif has very simple regulation – a single transcription factor. SPI-1 has extraordinarily complex regulation and the differences in approach were around figuring out what aspects are important and replicating them with synthetic regulation. We discuss this extensively in the paper, including expression tuning for maintaining stoichiometry of T3SS complex components, incorporation of core parts, and controlling systemic balance.